# Silica Nanospheres Coated Silver Islands as an Effective Opto-Plasmonic SERS Active Platform for Rapid and Sensitive Detection of Prostate Cancer Biomarkers

**DOI:** 10.3390/molecules27227821

**Published:** 2022-11-13

**Authors:** Anamika Pandey, Subhankar Sarkar, Sumit Kumar Pandey, Anchal Srivastava

**Affiliations:** 1Department of Physics, Institute of Science, Banaras Hindu University, Varanasi 221005, India; 2School of Computational and Integrative Sciences, Jawaharlal Nehru University, New Delhi 110067, India

**Keywords:** silica nanosphere, SERS, sarcosine, silver islands, WGM resonance, photonic nano-jets

## Abstract

The in vitro diagnostics of cancer are not represented well yet, but the need for early-stage detection is undeniable. In recent decades, surface-enhanced Raman spectroscopy (SERS) has emerged as an efficient, adaptable, and unique technique for the detection of cancer molecules in their early stages. Herein, we demonstrate an opto-plasmonic hybrid structure for sensitive detection of the prostate cancer biomarker sarcosine using silica nanospheres coated silver nano-islands as a facile and efficient SERS active substrate. The SERS active platform has been developed via thin (5–15 nm) deposition of silver islands using a simple and cost-effective Radio Frequency (RF) sputtering technique followed by the synthesis and decoration of silica nanospheres (~500 nm) synthesized via Stober’s method. It is anticipated that the coupling of Whispering Gallery Modes and photonic nano-jets in SiO_2_ nanospheres induce Localized Surface Plasmon Resonance (LSPR) in Ag nano-islands, which is responsible for the SERS enhancement. The as-fabricated SERS active platform shows a linear response in the physiological range (10 nM to 100 μM) and an extremely low limit of detection (LOD) of 1.76 nM with a correlation coefficient of 0.98 and enhancement factor ~2 × 10^7^. The findings suggest that our fabricated SERS platform could be potentially used for the rapid detection of bio-chemical traces with high sensitivity.

## 1. Introduction

The quick and accurate detection of cancer markers is a vital aspect of early diagnosis and successful treatment of cancer, as it has become one of the leading causes of death of humans globally [1]. Sarcosine, known as N-methylglycine, is an amino acid derivative and one of the cancer biomarkers, which has been extensively used for diagnostic purposes. It has 13 atoms pertaining to 33 normal modes of vibrations, which include 23 in-plane normal mode vibrations and 10 out-of-plane vibrational modes. All the vibrational modes of sarcosine are Raman active and belong to C_1_ point group symmetry [2]. In normal human blood serum, the average concentration of sarcosine is 1.4 ± 0.6 μM [3], and increased concentration of sarcosine is a potential marker of prostate tumors [4,5]. At an early stage, this growing tumor does not cause pain and does not show any frightening signs as it grows very slowly and later causes death [6,7].

During the 19th century, prostate cancer (PCa)was a silent and rare disease, but according to the report of the international agency for cancer research, it has become very common in the 21st century and has become the fourth among all cancers [8]. According to this survey, PCa accounts for about 7.1% of all cancer cases and around 3.8% of all prostate cancer deaths. Therefore, early diagnosis of this PCa is of utmost importance to reduce mortality and increase the survival rate of the patients. Until now, there have been various methods for screening for PCa biomarkers, such as computed tomography (CT), MRI/TRUS fusion guided biopsies, prostate-specific antigen (PSA) test, NMR profiling of metabolites, etc. [2,3]. However, CT and MRI/TRUS are not capable of the early-stage detection of PCa, whereas PSA is prostate-specific, not cancer-specific; hence, the use of PSA also has its limitations, such as organ-specific, small quantities present in blood serum of healthy men, altered level due to infection or another benign prostatic hyperplasia (BPH), and prostatitis or diet alteration which results in false positives [9]. All these traditional methods are tedious, costly, and time-consuming and fail to detect cancer in its early stages or at very low concentrations of PCa biomarkers. Therefore, we need non-invasive, economical, reliable, easy-to-use diagnostic tools, which can be used for early-stage PCa biomarkers and accurate in vitro diagnosis to cure PCa [10,11].

Raman spectroscopy has proven to be an indispensable technique for bio-chemical traces based on the vibrational fingerprint of molecules and also provides information about the compositions and structure of the molecules [12,13,14]. However, when the concentration of the target analyte is low, it shows weak vibrational bands. As a result, this technique is unable to detect the molecules because of the lower scattering cross section [15]. Later, resonant Raman spectroscopy and SERS were introduced to overcome this limitation of conventional Raman spectroscopy [16]. In resonant Raman spectroscopy, mostly excitation coincides with absorbance which produces a huge fluorescence in the background [17]. In SERS, these fluorescence signals are suppressed, and Raman signals are enhanced due to the amplification of electromagnetic fields created by the excitation of localized surface plasmons and allowing highly sensitive structural detection of analytes even at low concentrations [18]. This SERS activity can be explained by electromagnetic enhancement (EE) and chemical enhancement (CE). When these two mechanisms are combined, the SERS signal can be increased by ~10^14^ times [19]. SERS is a strong technology that gives ultrahigh sensitivity fingerprint vibrational information. However, only a few metals (gold, silver, and copper) produce a significant SERS effect, and they must be rough at the nanoscale, significantly limiting the range of SERS applications [20]. This technique has been extended to non-traditional substrates in a number of ways, the most notable of which is tip-enhanced Raman spectroscopy (TERS) in which the probed substance (molecule or material surface) can be on a generic substrate, and a nanoscale gold/silver tip above the substrate acts as the Raman signal amplifier. However, the total Raman scattering signal from the tip area is quite modest; hence TERS studies are limited to molecules with large Raman cross sections [21]. To overcome these limitations of SERS and TERS, Jian Feng Li et al. introduced a technique named shell-isolated nanoparticle-enhanced Raman spectroscopy (SHINERS) in which gold nanoparticles, synthesized by a standard sodium citrate reduction method, were coated with ultrathin silica or alumina shell under vigorous magnetic stirring. The authors reported the enhancement factor of the order of ~10^7^ [22]. Yan Hong et al. reported SERS detection of 3,4-methylenedioxymethamphetamine (MDMA) using the opto-plasmonic core-satellite microspheres surrounded with gold nanorods. The as-fabricated SERS substrate shows the limit of detection (LOD) of 5 nM [23]. In another report, Ying Yang et al. testified the detection of alpha-fetoprotein using Au@Ag@SiO_2_ as the SERS substrate. The fabricated SERS substrate shows linearity in the physiological range of 1 fg/mL to 1 ng/mL, having the limit of detection (LOD) ~0.3 fg/mL [1]. In the above reports, the authors used relatively tedious, costly, and time-consuming techniques to synthesize the metal nanoparticles and their composites, and sometimes due to the generation of undesired products or impurities, the performance of the materials also deteriorated [24,25]. In this context, RF sputtering has attracted the great attention of the scientific community, as it is one of the most feasible, easy to operate, scalable, and time-saving techniques to produce metal nanoparticles/nano-islands for SERS activity [26]. The main feature of this method is that it does not allow the generation of byproducts or impurities along with the thickness-controlled and homogeneous deposition of metal nano-islands [27,28].

In our work, we demonstrate an SiO_2_@Ag based SERS active platform for sensitive detection of the prostate cancer biomarker sarcosine using a very simple method. This facile and efficient SERS active platform has been developed via thin (5–15 nm) deposition of silver nano-islands using a simple and cost-effective RF sputtering technique followed by the synthesis and decoration of silica nanospheres (~500 nm) synthesized via Stober’s method. Under optimal conditions, the as-fabricated SERS active platform shows the linear response in the physiological range (10 nM to 100 μM) revealing an enormously low LOD of 1.76 nM, having a correlation coefficient of 0.98 and enhancement factor of 2 × 10^7^. The coupling of Whispering Gallery Modes (WGM), resonance modes, and photonic nano-jets in SiO_2_ nanospheres is expected to generate Localized Surface Plasmon Resonance (LSPR) in Ag nano-islands and is accountable for the enhanced Raman signal [29,30,31]. Thus, the findings suggest that our fabricated SERS platform could be a promising candidate for the early detection and better treatment of prostate cancer and also potentially be used for the rapid detection of other bio-chemical traces with high sensitivity.

## 2. Experimental

### 2.1. Chemicals

Tetra ethyl ortho-silicate (TEOS, purity ≥99% with trace metal basis) and hydrogen peroxide (H_2_O_2_) were received from Alfa Aesar. Ammonia solution (NH_4_OH), acetone (C_3_H_6_O), sulfuric acid (H_2_SO_4_), and ethanol (C_2_H_5_OH) were procured from Molychem, India. Sarcosine (C_3_H_7_NO_2_), used as a biomarker for PCa, was purchased from Sigma Aldrich Chemical Co, St. Louis, MO, USA. Glass coverslips were purchased from Polar Industrial Corporation, Mumbai, India. Deionized water (DI) was used as a solvent throughout the experiment. All the chemicals used were of analytical grade and used without any additional purification.

### 2.2. Instrumentation and Methods

The structural and morphological characterizations were performed using an X-ray diffractometer (PANalytical, Malvern, UK) with Cu-K_a_ radiation (α = 1.54178 Å) at a scanning rate of 1°/s ranging from 10° to 80°, scanning electron microscope (SEM) (ZEISS, Oberkochen, Germany) and an atomic force microscope (AFM) Park XE7 (Park, Suwon, South Korea). The photophysical and spectroscopic characterizations were carried out using Raman microscope (WITec alpha300 R|A|S, Ulm, Germany), Fourier transform infrared (FTIR) spectrometer (Frontier, PerkinElmer, Waltham, MA, USA), and UV-vis absorption spectrometer (PerkinElmer, Waltham, MA, USA).

#### 2.2.1. Synthesis of SiO_2_ Nanospheres

Silica nanospheres were synthesized via adopting the well-known Stober’s method as described in our earlier reported work [32] where tetra ethyl ortho-silicate (TEOS) was used as a precursor of silicon dioxide and ammonia solution as a catalyst as illustrated in Appendix A. In brief, 30 mL of ethanol (purity ≥98%) was taken in a beaker in which 6 mL of ammonia solution was added. This solution was stirred for 30 min. After that, 3 mL of TEOS was added which initiated a sol–gel reaction. This solution was kept for stirring for the next 14 h until the solution turned white in color. The milky solution was then washed with ethanol and then dried in a vacuum oven. The dried sample was grinned to obtain fine powder for further study.

#### 2.2.2. Fabrication of SERS Substrate

The schematic illustration of the fabrication of the SERS substrate is shown in Appendix A. To fabricate the SERS substrate, glass coverslip was used as the base material. The glass coverslip was properly cleaned with acetone and piranha solution (H_2_SO_4_ and H_2_O_2_ in a ratio of 3:1) to remove dust and other organic impurities and then rinsed with DI water. After cleaning the glass coverslip, an RF sputter coater (Lakshmi Vacuum Technologies PVT. LTD.) was used to grow silver nano-islands over the glass coverslip for 1 min at a constant power of 30 mW in the presence of argon gas (purity ≥99.95%, Linde India Limited) at ambient pressure of 0.5–0.6 Torr. The rate of silver deposition (0.1 Ås^−1^) on the substrate was constantly monitored with a digital thickness monitor associated with the coating unit. The slow rate of deposition was maintained to ensure uniform grain size. Furthermore, as-synthesized SiO_2_ nanoparticles were uniformly spin-coated over the silver nano-islands-deposited glass coverslip and dried for a minute. These silver glass coverslips coated with SiO_2_ nanospheres were used as SERS substrates for further investigations.

#### 2.2.3. SERS Measurements

The room temperature Raman and SERS spectra were recorded using a WITec alpha300 R|A|S Raman spectrophotometer equipped with a 532 nm solid-state laser. A short-distance objective lens, Zeiss Epiplan, of 50× with 0.65 NA was used to focus the incident laser beam on the sample and to collect the Raman-scattered signal with a grating of 600 lines per mm. Raman spectra were obtained in a single scan with an integration time of 10 s, and the laser power calibrated on the sample was 2.5 mW. The typical measured spectral resolution of our Raman spectrometer was found to be ~2.8 cm^−1^. Data processing was performed using project five 5.3 software. Furthermore, SERS spectra were obtained after spin coating a drop (50 mL) of different concentrations of sarcosine solution (10^−4^ M–10^−8^ M) to the as-fabricated SERS substrate. The overall detection procedure for prostate cancer biomarker, viz., sarcosine at different concentrations is illustrated in Appendix A.

Briefly, the known concentrations of sarcosine were drop-casted over the fabricated SERS substrate. Subsequently, the sarcosine was attached with silica nanoparticles. We propose that sarcosine is adsorbed in a negatively charged silica surface with its positively charged amino group in its zwitterion form via electrostatic interaction as shown in Figure 1. Furthermore, this substrate was subjected to laser illumination for SERS measurements. The scattered signals were detected by a silicon-based CCD detector of the Raman microscope.

## 3. Results and Discussion

### 3.1. Characterization of As-Synthesized SiO_2_ Nanoparticles

Figure 2a,b show the SEM images of as-synthesized SiO_2_ nanoparticles at different magnifications, clearly indicating that the bulk formation of good quality sphere-like structures has average lateral dimensions of ~500 nm.

Furthermore, the structure of the as-synthesized sample was confirmed by X-ray powder diffraction using an X-ray diffractometer. Figure 2c shows the typical XRD pattern of the as-synthesized sample. The pattern displayed a broad diffraction peak around 2θ = 23.3° which corresponds to the plane (101) and was successfully indexed with JCPDS Card No. 47-0715. This result indicates that the as-synthesized sample is amorphous in nature.

To visualize the thickness and surface topography of the as-synthesized SiO_2_ sample, AFM measurements were carried out in non-contact mode. The 2D topographical image of SiO_2_ nanoparticles is represented in Figure 2d. This clearly indicates that the synthesized samples possess spherical shapes having an average height of ~500 nm as shown in Figure 2e.

It is well-known that the Raman spectroscopy is a versatile technique that deals with the purity, defects, and structural properties present in the materials [33]. Raman spectra of the synthesized samples were recorded using WITec alpha300 R|A|S microscope having a laser excitation wavelength of 532 nm. As shown in Figure 2f, the recorded spectrum consists of two prominent peaks observed at 877 cm^−1^ and 974 cm^−1^. The peak found in the region 740 to 880 cm^−1^ could be due to the Si-O-Si bending mode, and the peak obtained at 974 cm^−1^ is attributed to the stretching vibration of surface Si-(OH) bonds [34,35].

Furthermore, to investigate the functional validity and identify the various functional groups introduced over the synthesized samples, the FTIR spectrum of the synthesized SiO_2_ nanospheres was recorded. As shown in Figure 2g, the recorded spectrum has several intense absorption bands around ~800 cm^−1^, 945 cm^−1^, and 1062 cm^−1^ which correspond to the characteristics bands of SiO_2_ such as Si-O-Si bending mode, Si-OH stretching vibration, and antisymmetric Si-O-Si stretching vibration, respectively. Along with these, the spectrum also possesses two more bands around 1632 cm^−1^ and 3406 cm^−1^, corresponding to the O-H bending and OH-stretching vibration modes of physisorbed water molecules. The obtained results are well-matched with earlier reported works on SiO_2_ [36,37].

Appendix A shows the UV-vis absorption spectra of synthesized SiO_2_ nanospheres. To know the band gap of the as-synthesized SiO_2_ nanosphere, the UV-Vis spectrum was recorded. From the figure, it can be clearly seen that a single characteristic absorption band at ~261 nm has been observed due to the nonparamagnetic defects in silica, referred to as the B_2_ band, attributed to occur with some kind of oxygen deficiency in the silica network [38]. Furthermore, we also calculated the optical band gap of SiO_2_ nanoparticles with the help of Tauc’s plot which is shown in Appendix A. We found that the figure shows one absorption edge corresponding to a band gap of about 3.7 eV.

### 3.2. AFM analysis of Deposited Silver Nano-Islands

The topographical images and the corresponding height profiles of the as-grown silver nano-islands and silica-decorated silver nano-islands on the glass coverslip were recorded using the atomic force microscopy (AFM) technique, employing the atomic force microscope instrument Park XE7 (Park, Suwon, South Korea) in non-contact mode. Figure 3a shows the 2D AFM topography of the silver nano-island deposited over the glass coverslip via the RF sputtering technique, clearly indicating the uniform and conformal layer of silver nano-islands deposited over the glass coverslip. The inset of Figure 3a shows the height profile of nano-islands across the red line depicting the average height of grown nano-islands around 5–15 nm. Figure 3b is the schematic representation of the deposited silver nano-islands over the glass coverslip. Figure 3c shows the 2D AFM topography of silica nanospheres-decorated silver nano-islands over the glass coverslip. The inset of Figure 3c shows the height profile of nanospheres across the red line depicting the average height of grown nanospheres around 500 nm. Figure 3d is the schematic representation of the silica nanospheres-decorated silver nano-islands deposited over the glass coverslip. The 3D AFM topography of silver nano-islands and silica nanospheres-decorated silver nano-islands is shown in Appendix A.

Appendix A show the recorded Raman spectrum of pure bulk sarcosine powder in full range (50 cm^−1^ to 3200 cm^−1^) and in a specific range (850 cm^−1^ to 1750 cm^−1^), respectively. It can be clearly seen that the recorded spectrum consists of a number of sharp and intense characteristic peaks around 3024 cm^−1^, 2993 cm^−1^, 2957 cm^−1^, 1633 cm^−1^, 1594 cm^−1^, 1486 cm^−1^, 1466 cm^−1^, 1418 cm^−1^, 1384 cm^−1^, 1290 cm^−1^, 1173 cm^−1^, 1057 cm^−1^, 972 cm^−1^, 936 cm^−1^, 893 cm^−1^, and 366 cm^−1^, etc., along with other low intense peaks corresponding to various Raman active vibrations, which are listed in Appendix A. This is in good agreement with the previously reported works on sarcosine biomarker [2,39,40].

Figure 4a shows the Raman spectra of different concentrations of sarcosine on a bare glass coverslip. It is clear from the obtained spectra that no significant Raman peak has been detected in this physiological region. Furthermore, the Raman spectra of sarcosine of different concentrations have been recorded directly on the Ag nano-islands without coating as-synthesized silica nanospheres, as shown in Figure 4b. The figure clearly shows that it is very hard to detect the sarcosine peaks in this physiological range, as the peak enhancement is very low. Although we have increased the concentration of sarcosine up to 10^−4^ M, the intensity of the Raman signals has not significantly increased with respect to the increasing concentration.

Furthermore, to examine the SERS enhancement, the Raman spectra of sarcosine at concentration 10^−6^ M were recorded over the bare Ag and SiO_2_@Ag platform and compared with bulk sarcosine powder, as shown in Figure 4c. Interestingly, from the comparative plot of Raman spectra for pure bulk sarcosine powder (blue line), 10^−6^ M concentration of sarcosine on bare Ag as a SERS substrate (red line), and silica-modified SERS substrate (black line the huge enhancement in the SERS signal of sarcosine, when SiO_2_ was used as a SERS substrate can be clearly observed. Thus, the SERS enhancement in the Raman peak at 1594 cm^−1^ helps us identify the trace amount of sarcosine in prostate cancer samples at its early stage. Moreover, if we compare the Raman spectra of sarcosine over the silica-coated Ag (SiO_2_@Ag) substrate with the Raman spectra of pure sarcosine powder, then we notice some minor shifts in the peak position. It can be because of two factors: (i) absorption of sarcosine molecules over the surface of the SERS substrate and (ii) intermolecular interaction with water in an aqueous solution. Accordingly, the as-prepared nano-islands with an amiable uniform silica layer may also be promising in the relatively quantitative detection of biomolecules with the advantages of simplicity and sensitivity.

Furthermore, to understand the behavior of the as-fabricated SERS substrate (SiO_2_@Ag) with different concentrations of sarcosine, the Raman spectra were recorded in the physiological range (10^−4^ M to 10^−8^ M), as depicted in Figure 4d. From the figure, it can be clearly seen that over the silica-coated Ag (SiO_2_@Ag) substrate Raman spectra of sarcosine show a huge enhancement in its signal, which is due to the amplification of electromagnetic fields created by the excitation of localized surface plasmons. That allows highly sensitive detection of sarcosine in the physiological range (10^−4^ M–10^−8^ M). The Raman peaks of 10^−4^ M solutions of sarcosine are much higher than the pure sarcosine having the same concentration. The SERS response of sarcosine molecules on the as-prepared SERS substrate exhibits a proportional decay behaviour as the concentration of the sarcosine molecule decreases.

The Raman spectrum of sarcosine shows a Raman peak around 2993 cm^−1^ which was merged in SERS spectra and shows a broad band with relatively low intensity at 2957 cm^−1^ corresponding to symmetric stretching vibrations of CH_2_. The reason behind this is that the SERS active substrate absorbed most of the CH_2_ groups resulting in a reduction in the Raman signals obtained due to the C-H stretching vibrations. In general, the Raman band corresponds to the asymmetric COO^−^ stretching vibrational modes that occur around 1600–1570 cm^−1^. The COO^−^ stretching mode was coupled with the deformation mode of NH_2_. Both the shrinkage of the donor–acceptor distance and the strengthening of the N-H∙∙∙O hydrogen bonds can be used to explain these simultaneous changes in the two modes and show a huge enhancement in the SERS spectra. This is due to the COO^−^ group being normally aligned with the fabricated SiO_2_@Ag SERS substrate, while the reason behind the broadening of the Raman band is the coupling of the COO^−^ stretching vibrational mode with the C=O stretching mode and NH_2_ deformation mode [2,39,41].

Appendix A shows the optical images of the film before and after the Raman measurements and from the optical images. We found that there is no change in its morphology. We also recorded the Raman spectrum of as-synthesized SERS substrate (SiO_2_@Ag) which is shown in Appendix A.

The enhancement factor (EF) is an important parameter to compute the enhancement ability of any SERS substrate. Thus, the EF of the sarcosine adsorbed on the surface of the SiO_2_@Ag platform was estimated using Equation (1), which is also in good agreement with the previously reported works, given below [19,42,43];
EF = (I_SERS_ × N_NRS_)/(I_NRS_ × N_SERS_)(1)
where I_SERS_ and I_NRS_ are the SERS intensity and normal Raman intensity, respectively, for the specific peak position. N_SERS_ and N_NRS_ are the number of sarcosine molecules contributing to the SERS signal and the Raman signal, respectively. Different parameters, including Avogadro’s number (6.022 × 10^23^), the size of the glass coverslip used to make the SERS substrate, the concentration of the Raman marker employed, and the radius of the laser point, were utilized to determine N_SERS_ and N_NRS_. Hence the above Equation (1) can be modified as:EF = (I_SERS_ × C_NRS_)/(I_NRS_ × C_SERS_)(2)

In the case of the SiO_2_@Ag substrate, the typical measured value of I_SERS_ is found to be ~50 (a.u.) for the C_SERS_ of 10^−8^ M, and for bare glass coverslip, the typical measured I_NRS_ is ~25 (a.u.) for C_NRS_ of 10^−1^ M. However, for Ag-deposited glass coverslip, the obtained I_SERS_ is ~50 (a.u.) for C_SERS_ of 10^−4^ M. By substituting the values in the above Equation (2), the EF for the SiO_2_@Ag platform was estimated as ~2 × 10^7^. However, in the case of Ag-deposited glass coverslip, the EF was found to be ~2 × 10^3^. This suggests that the as-fabricated SERS substrate (SiO_2_@Ag) has better EF than the Ag-deposited glass coverslip, i.e., more than 10^4^ times, reflecting its potential utility in the rapid detection of bio-chemical traces with high sensitivity. Furthermore, we can say that the method described by Li et al. is relatively tedious and time-consuming as it has a two-step synthesis technique. Sometimes, in the process of synthesizing the metal nanoparticles as well as their composites, the production of some undesired products or impurities deteriorates the performance of the materials. However, in our reported work, we used a simple RF sputtering method for the deposition of Ag nano-islands along with the coating of silica nanospheres over it by a spin coating method, and we obtain equal EF to the work reported by Li et al. Thus, we believe that our reported technique is much more effective compared to the work reported by Li et al. that uses the SHINERS method to obtain the EF of order ~10^7^ [22].

### 3.3. Calibration Curve

The calibration curve obtained between the concentration of sarcosine and the corresponding intensity is shown in Figure 5. We chose the Raman peak of sarcosine that occurred at 1594 cm^−1^ as the analytical band, and we can observe that the peak intensity is proportional to the logarithm of sarcosine concentrations in the physiological range (10 nM to 10^5^ nM). The SERS response of the fabricated SiO_2_@Ag SERS substrate at the different concentrations of sarcosine revealed a correlation coefficient of 0.98. The LOD of the fabricated SERS substrate was calculated using Equation (3) given below [44];
LOD = 3.3 × (SD of intercept/slope)(3)

By substituting the values of SD of the intercept and the slope in the above Equation (3), the LOD is found to be 1.76 nM. (The detailed calculation of the LOD is described in the Appendix A). Here, in our case, we anticipated that the coupling of Whispering Gallery Modes and photonic nano-jets in SiO_2_ nanospheres induce Localized Surface Plasmon Resonance (LSPR) in Ag nano-islands, which is responsible for the SERS enhancement. Moreover, SiO_2_ nanospheres act as nano-lens, so it is able to detect a very small amount of biomarker; consequently, the LOD is ultralow.

### 3.4. Understanding the Role of the SiO_2_@Ag Substrate in the Enhancement of the SERS Signal

Among all the noble metals, Ag has been considered as the most attractive plasmonic contender for SERS applications because of its low cost, relatively low optical frequency loss, and high plasmonic efficiency than other noble metals. However, due to the high surface activity, Ag nanoparticles are prone to contamination, oxidation, and agglomeration in biological samples. As a result, utilizing them as a SERS substrate for biological samples is exceedingly challenging. In this respect, so many efforts have already been made to improve the chemical stability of Ag NPs [22,45], in which the core-shell nanostructures or cladding or decoration with various types of nanostructured materials such as graphene, reduced graphene oxides, Cu_2_O, TiO_2_, ZnS, AlO_3_, SiO_2_ nanospheres, etc., are the most popular techniques for protecting Ag NPs from oxidation, agglomerations and long-term stability, especially in moist environments. Herein, the thin decoration or cladding of a non-metallic, chemically inert, transparent, hydrophilic, and biocompatible SiO_2_ surface serves as a protective covering for the Ag nanoparticles and increases Raman intensity to some extent [22]. In our experiments, the coating of SiO_2_ nanospheres over the deposited Ag nano-islands protected the Ag layer from possible contaminations and also served as a closed-pack array of nano-lenses to enhance the SERS intensity via various resonant and non-resonant enhancement mechanisms, such as Mie-type scattering, Whispering Gallery Modes (WGM), optical trapping, and photonic nano-jets. On the other hand, state-of-the-art plasmonic metal surfaces, such as Ag nano-islands, which are rough at the nanoscale, can confine the electromagnetic fields in nanoscale regions called “Electromagnetic Hotspots”. Analyte molecules that are close to those electromagnetic hotspots can scatter the light, causing an enhanced SERS signal with an EF of several orders of magnitude. Here, combining Ag nano-islands as a plasmonic base and SiO_2_ spheres as nano-lenses results in an opto-plasmonic hybrid structure, which exploits coupling between both the photonic and plasmonic enhancement mechanisms to produce a much more enhanced SERS signal than the individual enhancement mechanisms can produce alone [46].

There are many ways reported in the literature through which nano-lenses can enhance the intensity of incident radiation. Dielectric nano-lenses can focus light into a sub-wavelength region and can generate high-intensity “photonic nano-jets” of a few hundred nanometer widths which emerge from the shadow side surfaces of the spheres [47,48]. As shown in Figure 6a, these photonic nano-jets can significantly induce the LSPR into the Ag nano-islands under them, forming high-intensity electromagnetic hotspots which further enhance the Raman scattering by the analyte molecules, causing an enhanced SERS intensity with several orders of magnitude. 

Sufficiently large and high refractive index dielectric lenses can produce evanescent fields through total internal reflection, which travels through the surface of the sphere as is called WGM resonance. As shown in Figure 6b, WGM waves scan the analyte molecules adsorbed in the sphere surface several times, resulting in an increased SERS intensity [49,50].

In our case, one more enhancement mechanism seems feasible. The SiO_2_ nanospheres, having a higher refractive index of 1.46 compared to surrounding air (refractive index = 1.00), can trap the incident light through the exited WGM modes due to their spherical shape [51]. When the light inside the nanosphere incidents at an angle greater than 43^0,^ which is the critical angle of the SiO_2_-air interface, the light suffers several total internal reflections and propagates close to the sphere surface and becomes trapped. When light completes one round-trip around the sphere surface, it may be in phase with the initial position, which produces sustained WGM modes under the condition,
2πr = *l* λ/n(4)
where λ is the wavelength of the incident light in vacuum, n is the refractive index of the dielectric medium, and *l* is the number of wavelengths present in a complete round-trip along the sphere surface. The higher the number *l*, the more sustained the resonant modes will be and the more capacity of optical trapping will be [51]. In our case, the incident wavelength of the laser beam is 532 nm, and the refractive index is n = 1.46. So, the SiO_2_ nanospheres may be able to guide the light around their surfaces via resonant WGM modes, and there will be ~eight wavelengths in a complete round-trip around the SiO_2_ spheres. These resonant modes can couple the light with the Ag nano-islands to induce the LSPR effect to further enhance the SERS intensity [52]. However, we believe that some other studies, for example, FDTD simulation-based studies, may also be carried out for further validation of the same, which we will try to incorporate into our future studies.

To enrich our manuscript, we thoroughly investigated the related works and made a comprehensive comparison of our work with these works which is presented in Table 1.

Therefore, the SERS active platform fabricated in this work shows a better physiological range (10 nM to 100 µM) of detection with an excellent LOD of 1.76 nM, which is better than or comparable to other reported works, indicating that our fabricated SERS substrate is extremely sensitive and has an ultralow limit of detection.

## 4. Conclusions

In summary, we fabricated a facile, scalable, and efficient SERS platform for the sensitive and rapid detection of the sarcosine prostate cancer biomarker. A simplistic one-hand synthesis of silica nanoparticles was performed using Stober’s method. The as-synthesized silica nanoparticles were well characterized by various characterization techniques such as SEM, AFM, Raman, XRD, UV-vis, and FTIR, which shows the bulk formation of good quality sphere-like amorphous structure of silica nanoparticles. Furthermore, a SERS active platform was developed via thin (5–15 nm) deposition of silver islands using a simple, cost-effective, and thickness-controlled RF sputtering technique followed by the uniform decoration of as-synthesized silica nanospheres (~500 nm). The benefit of decorating SiO_2_ nanospheres is that it protects the Ag islands as well as improves the enhancement of the electromagnetic field via coupling of WGM resonance modes and photonic nano-jets in SiO_2_ nanospheres with induced LSPR in Ag nano-islands. Under the optimal condition, the as-fabricated SERS active platform shows a linear response in the physiological range (10 nM to 100 μM) and an enormously low LOD of 1.76 nM with a correlation coefficient of 0.98 and an enhancement factor ~2 × 10^7^. Thus, the findings suggest that our fabricated SERS platform could be a promising candidate for the early detection and better treatment of prostate cancer and also potentially be used for the rapid detection of other bio-chemical traces with high sensitivity.

## Figures and Tables

**Figure 1 molecules-27-07821-f001:**
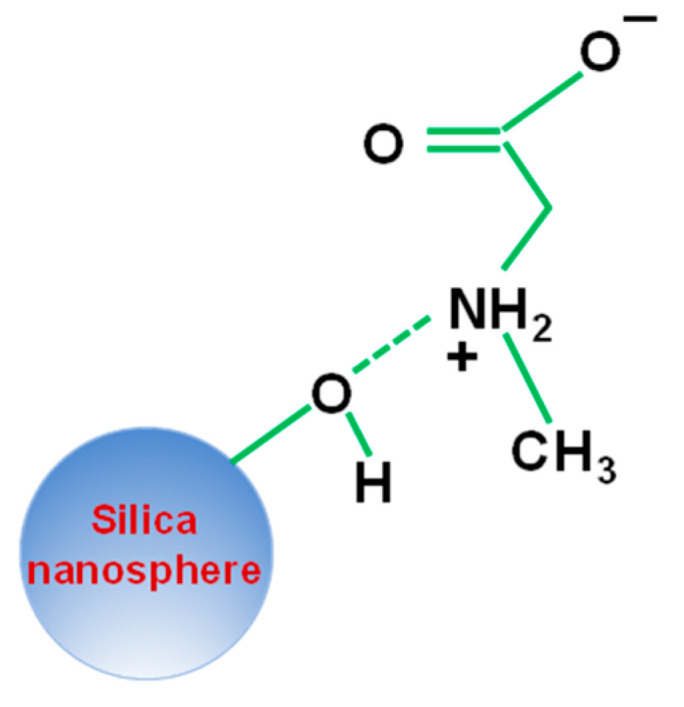
Schematic illustration of the attachment of the sarcosine cancer biomarker with silica nanoparticles.

**Figure 2 molecules-27-07821-f002:**
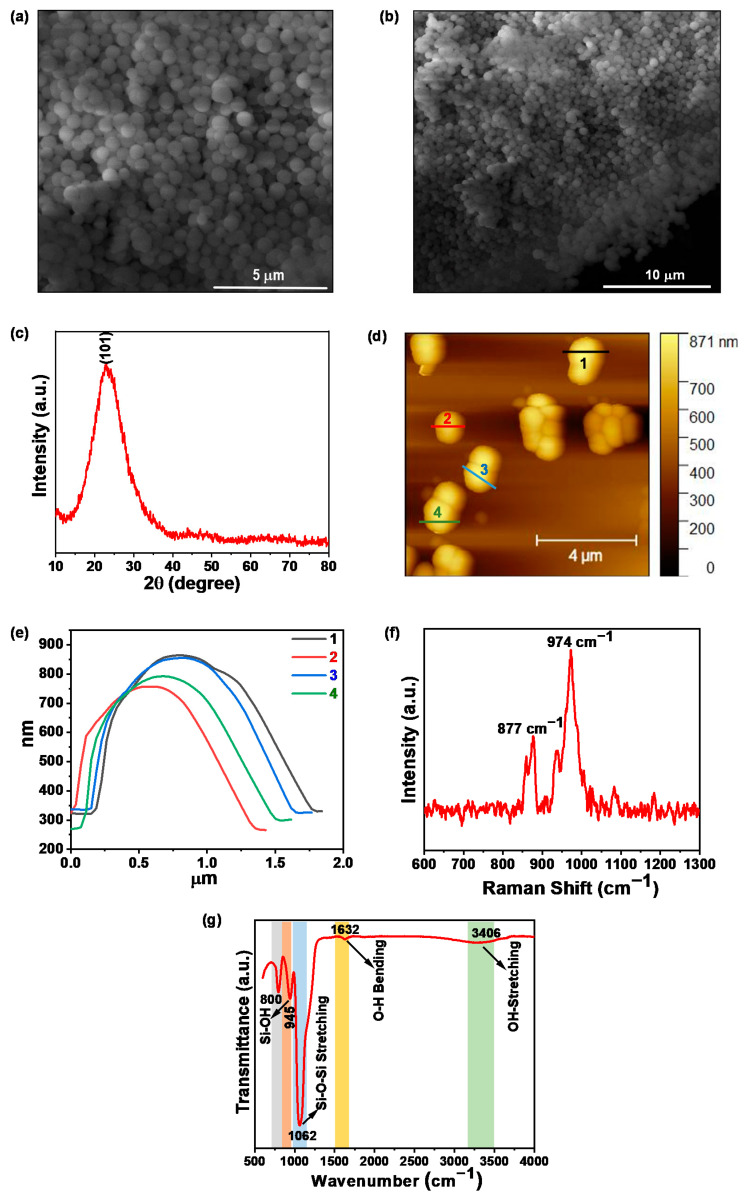
(**a**,**b**) SEM micrographs of SiO_2_ nanoparticles at different magnifications; (**c**) typical XRD pattern of as-synthesized SiO_2_ nanoparticles; (**d**) 2D topography of as-synthesized SiO_2_ nanoparticles; (**e**) height profile of SiO_2_ nanoparticles along the color-coded lines depicting the average size of nanoparticles ~500 nm; (**f**) Raman spectrum of as-synthesized SiO_2_ nanospheres; (**g**) recorded FTIR spectrum of SiO_2_ nanospheres.

**Figure 3 molecules-27-07821-f003:**
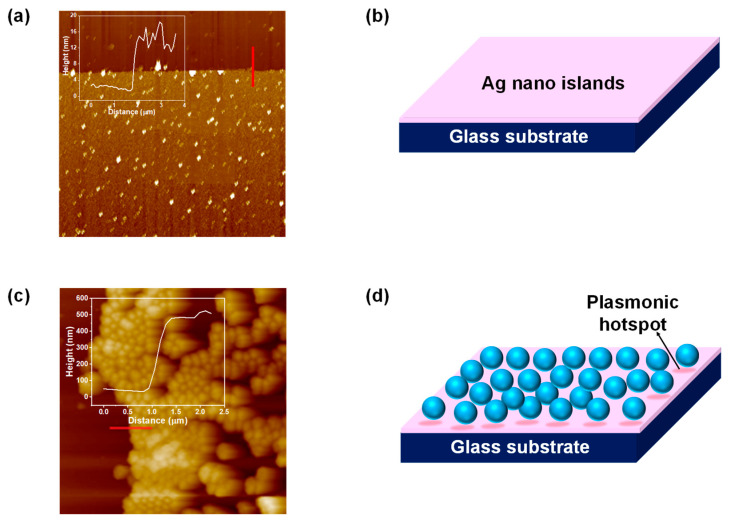
(**a**) 2D topography of silver nano-islands deposited over glass coverslip and the height profile (inset) along the red line of the surface; (**b**) schematic representation of silver nano-islands deposited over glass coverslip; (**c**) 2D topography of silica nanospheres-decorated silver nano-islands deposited over glass coverslip and the height profile (inset) along the red line of the surface; (**d**) schematic representation of silica nanospheres-decorated silver nano-islands deposited over glass coverslip.

**Figure 4 molecules-27-07821-f004:**
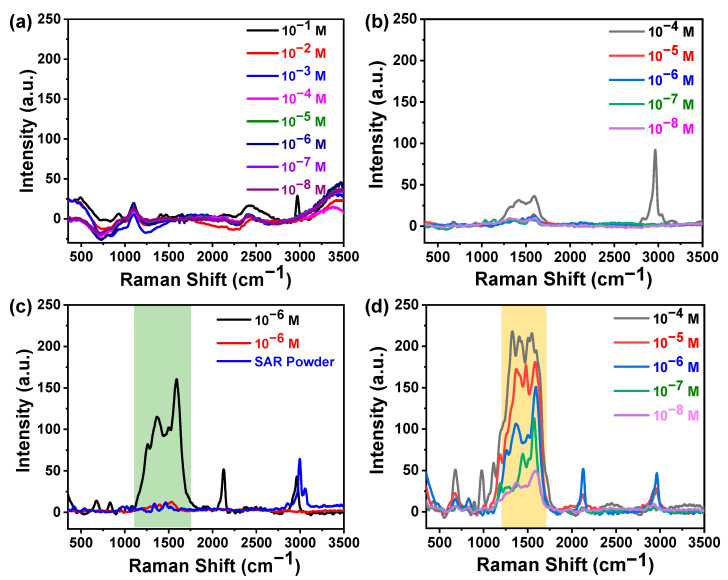
(**a**) Recorded Raman spectra of sarcosine at different concentrations; (**b**) recorded Raman spectra for different concentrations of sarcosine deposited over bare Ag as a substrate, indicating that no significant SERS enhancement was observed; (**c**) a comparative plot of obtained Raman signals of pure bulk sarcosine powder (blue line), 10^−6^ M concentration of sarcosine over bare Ag as a SERS substrate (red line), and silica modified SERS substrate (black line); (**d**) recorded Raman spectra for different concentrations of sarcosine over the SiO_2_ decorated silver nano-islands as a SERS substrate, indicating the significant enhancement of Raman signals even at very low concentration of sarcosine ~10^−8^ M.

**Figure 5 molecules-27-07821-f005:**
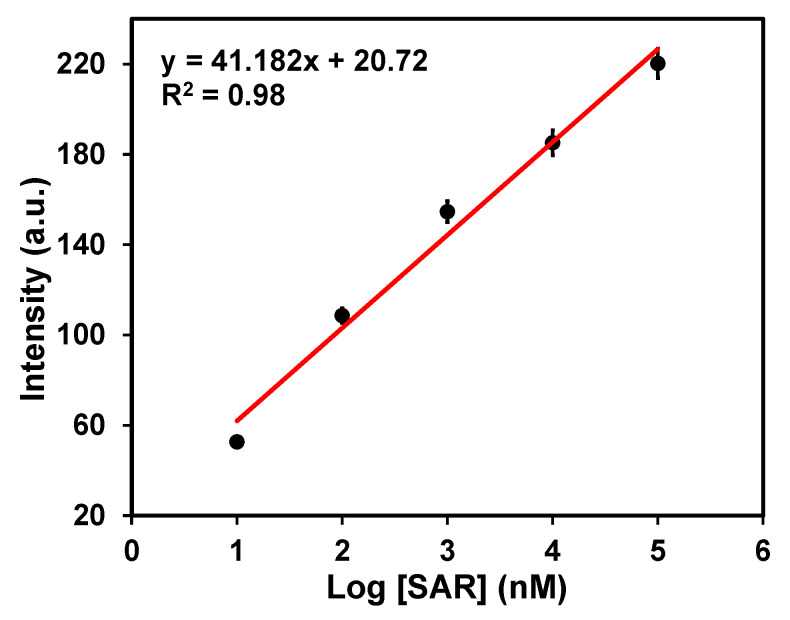
Calibration curve between intensity recorded with different concentrations of sarcosine.

**Figure 6 molecules-27-07821-f006:**
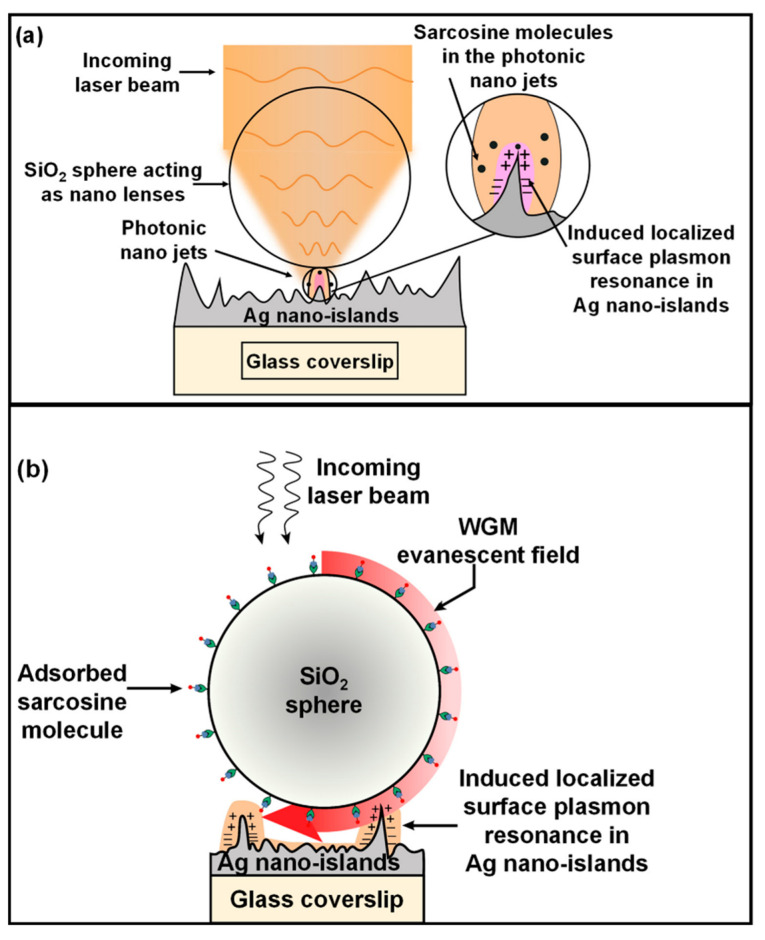
(**a**) The incoming laser beam becomes focused by the SiO_2_ microspheres creating a high-intensity subwavelength width photonic nano-jet that scans the sarcosine molecules inside it. Photonic nano-jets induce LSPR in Ag nano-islands to further increase the local field intensity which further enhances the SERS signal; (**b**) the resonant WGM waves propagate close to the SiO_2_ sphere surface, scanning the sarcosine molecules adsorbed in it. It further induces LSPR into the Ag nano-islands, increasing the local field which causes SERS enhancement.

**Table 1 molecules-27-07821-t001:** Comparison of the present work with other reported works in which SERS platforms were fabricated using different nanomaterials.

S. No.	Technique	Material	Detection Range	Limit of Detection	Reference
1.	Fluorescence	Au NPs	100 nM–50 mM	10 nM	[53]
2.	icELISA	Acrylamide-4-MBA	124.79 nM–1590 nM	124.79 nM	[54]
3.	MIP-DPV	Au NPs-MWCNTs	703.4 nM–70.34 μM	393.9 nM	[55]
4.	FIC	Luminol-Hydrogen Peroxide	1.0 μM–1.0 mM	39.5 nM	[56]
5.	SWV	Au NPs	2.0 μM–50 mM	158 nM	[57]
6.	SERS	Silica coated Ag nano-islands	10 nM–100 μM	1.76 nM	Present work

## Data Availability

The data presented in this study are available on reasonable request from the corresponding author.

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
