# Peer review of "Silica Nanospheres Coated Silver Islands as an Effective Opto-Plasmonic SERS Active Platform for Rapid and Sensitive Detection of Prostate Cancer Biomarkers"

_molecules, 2022, doi:10.3390/molecules27227821_

Round 1

Reviewer 1 Report

The paper considers the detection of sarcosine at low concentrations using the SERS technique based on silver nanoislands coupled with SiO2 nanospheres. The detection of sarcosine is of great importance in terms of the early stage prostate cancer diagnosis, that determines the high actuality of the paper. The manuscript is well structured and easy to read. The authors use various experimental techniques to characterize the structures under study. The results are justified and well presented.

I think the manuscript can be published as it is. I have just one small note: perhaps the role of the SiO2 nanospheres is to attach the sarcosine molecule, which is why the biomarkers are not evenly distributed in solution, but relatively close to the SERS substrate, and this is why the signal is amplified so much.

Reviewer 2 Report

The paper is devoted to the fabrication of silver nanoisland film decorated with silica microspheres and probing SERS properties of the fabricated substrate in chemosensing of amino acid sarcosine (potential cancer marker).

Putting aside the fact that sarcosine is a highly debatable cancer marker (primarily because of its high and variable normal levels), the motivation for the is relevant; and fabrication, characterization and SERS experiments are fairly well done.

However, the description and explanation of the results have some major issues, listed below

 1. Please give more details about the application of sarcosine (Line 235-…) to the film. Was it also dissolved in deionized water? Was it dried? If yes, how did you control the amount of residual water and spot size? Was Marangoni (“coffee spot”) effect observed and accounted for? If no, how did you prevent evaporation during measurement? How did you control beam focus in the liquid?

2. Proof is needed that the observed band at (figure 5d) are not bands of amorphous carbon (“charcoal”, 1400 & 1600 nm), which can be formed by pyrolysis of amino acids in highly focused light field (even under liquid water, personal experience). Did you examine the film microscopically after the Raman experiment? Did you study spectrum dependence on laser power?

3.  (Line 242 and after, Figure 4) Although this figure can be found in some photophysical papers, the displayed direction of this reaction is not feasible in water. On the contary, the silyl ester will decompose and release carboxylic group even with traces of water vapour (example - curing of acetoxy silicone gels). The displayed reaction would be possible only with carboxyanhydrides or in the presence of strong water-removing agents. Moreover, if amino acids reacted with SiO2 surface this way, half of biochemical studies and methods would be invalid. It may be speculated that even the non-covalent interaction of silica surface (acidic and negatively charged in aqueos medium) with negatively charged carboxylic group of sarcosine is electrostatically unfavourable. Please provide more feasible explanation of sarcosine/glass interaction.

4. Pages 9 and 10, calibration curve: if we look at figure 6b we see a linear dependence of signal on the logarithm of concentration. So line 311, which states ‘proportional to the concentration’ is not correct. Actually, the logarithmic dependence of Raman signal on concentration needs detailed explanation, as it is not obvious from the physics of the process. If we use equation (2) for different concentrations we will have different enhancement factors? Why? Also, there are other peaks outside the 1300-1700 range in the enhanced spectrum (Figure 6a), which show different and even non-monotonous dependence on concnetration. Are there any explanations for this?

5. (Figure 7a, Lines 380 and after) As the silica microsphere is a Mie scatterer (raduis~wavelength) the geometic optics is inapplicable to it. The microspheres should also have near field interaction with glass substrate and with each other (as judged from AFM images). I have not found if Maxwell equation solutions are available for this geometry, but the rays in Figure 7a are very misleading.

 Please also consider the following: your study has demonstrated high sensitivity of the potential sensor, but told us nothing about its selectivity. Will it be able to firmly differentiate sarcosine from glycine? From acetic acid? The potential interference from ions the solution was not addressed either. Unfortunately, most studies of potential biosensors do not. High sensitivity in deionized water is good, but purification of sarcosine from biological samples and its deep desalination would be very expensive and strongly overweight the cost efficiency of the sensor.

 Major revision of the manuscipt, which addresses the above points, is recommended.

Some minor points:

Line 34-35. Elevated sarcosine is a potential marker of prostate tumors, but is not a cause of tumors (it is an ubiquitous normal metabolite)

Line 105. Maybe “revealing” instead of “reviling”

Lines 107-108 Literature reference for WGM and LSPR is anticipated

The sentence in line 165 is probably excessive, as AFM is now a well-known mature method

Line 173 – grammar error?

Line 196. Are there proofs that this peak is not due to Mie resonance (λ=R)?

Table S1 - The displayed structure is not sarcosine, but 1-nitropropane (wrong position of nitrogen atom)

Reviewer 3 Report

The manuscript titled "Silica Nanospheres Coated Silver Islands as an Effective opto-plasmonic SERS Active Platform for Rapid and Sensitive Detection of Prostate Cancer Biomarkers" describes the preparation and characterization of a SiO2@Ag based SERS active platform for sensitive detection of prostate cancer biomarker sarcosine.

The work presented may be of interest to the readers of Molecules but some modifications need to be made before it can be considered for publication.

- RF sputtering: It is recommendable to use the full name (Radio Frequency Sputtering) at first (to explain the abbreviation before its first use).

- Pg. 2, lns. 61-62: In the resonance Raman spectroscopy an enhancement of the Raman signal instead of the fluorescence can be also observed. It is not clear from your explication.

- Pg. 2, lns. 77-90: It seems a little bit like a list of some achievements chosen just "for example" from the all ones found in the discussed field. I miss a guideline for this selection. In addition, there is no mention of the opto-plasmonic hybrid structures which is the principal topic of the Results and Discussion part.

- Pg. 4, lns. 149-155+ pg. 7, lns. 228-245 (+ Fig. 3): These parts should be moved to the Experimental part of the manuscript. In addition, Fig. 3 presents a common and routine procedure for sample addition and spectra recording. In my opinion, there is no need for this figure.

- Pg. 4, lns. 172-187 + Fig. 1f:  Why your Raman spectrum is depicted/discussed starting at 600 cm-1? Is there anything below 600 cm-1?

- Pg. 6, lns. 194-200: What conclusion regarding the obtained/characterized SiO2 nanospheres can be deduced from these observations?

- Fig. 2 b and d: These schematic representations do not have too much sense. Moreover, they can also be found in Fig. S2.

Besides, the quality of the figure (especially Fig. 2d) has to be improved.

- Pg. 7, ln. 232: Did you really use a grating of 600 lines per mm and a 532 nm-laser? What about your spectral resolution? Can you define it? Why did you not use a grating with an increased number of lines per mm?

- Pg. 7, lns. 242-243: This claim needs to be supported with a reference or in-depth discussion.

- Fig. 4 + Fig. 5: There is no need to have so large Fig. 4. On the contrary, the spectra depicted in Fig. 5 should be enlarged.

- Pg. 8, lns. 253-256: Some of the intense bands are not mentioned in this observed peaks-ring. Maybe, there is no need to mention so many wavenumbers, just to select the most intense ones. Moreover, Table S1 should list all observed Raman bands. Besides, we recommend adding a description of the relative intensity of each band.

Pg. 8, ln. 257: Can you also specify/discuss the marker bands in the Raman spectrum of sarcosine that could be defined as cancer markers/decisive in an early diagnosis of cancer?

Pg. 8, lns. 268-276: Can you specify from where - which structure - the COO- and NH2 vibrations/groups can be deduced?

Pg. 8, lns. 277-283: I recommend preparing a Figure where the Raman spectra of sarcosine recorded in its pure form, in a solution, on Ag- and SiO2@Ag-SERS substrates will be presented all in once.

Fig. 5b: Since the Raman spectrum of the sarcosine (powder) is known very well (has been already published), there is no need to present also the chosen specific range of the spectrum. Besides, all other spectra are depicted in the spectral range from 500 cm-1 to 3500 cm-1.

I miss blank spectra, e.g. the Raman spectrum of the prepared and used SiO2@Ag-SERS substrate.

I recommend reorganizing Fig. 5.  Firstly, c, e and f should be discussed, and just then, the concentration study can be presented.

- I miss deeper and longer discussions regarding the observed spectral changes.

- Are figures 5d and 6a the same, aren't they? There is no need for their repetition. Maybe you can remove the concentration study from Fig. 5 and present it just in Fig. 6.

- Pg. 10, Eq 1: What about the standard deviation of the response? Is there no need to count on anyone? Can you explain (more accurately) why your LOD is so ultralow when compared to others reported in the bibliography? It is really suspicious.

Pg. 10, lns. 323-338: I am just curious: Is not better to use for the EF calculation the intensity values obtained for the same concentrations? I am just thinking: by changing a concentration one can also observe some other effects, much more if you are comparing such different concentrations as 10-1 M and 10-8 M. 

Pg. 10, lns. 332-338: Your calculated EF is ~2×107. You wrote: "...the present technique is much more effective as compared to the work reported by Li et al. that uses the SHINERS method to get the EF of order ~107."

Why do you think your method is much more effective? Does the 2-times higher EF equal a much more effective method?

Pg. 11-12: Too descriptive and not supported by any robust data.

Pg. 13: Table 1 is not mentioned in the text of the manuscript.

Round 2

Reviewer 2 Report

The Authors have cleared most of the issues except one.

The graph in Figure 5 is still a logarithmic responce, not a linear responce. The intensity is proportional to logarithm of concentration, rather than to concentration value (no matter if you express it as actual numbers, or orders). Please remove the statement on linearity. Logarithmic dependence  is not an error, as many chemosensors (pH electrodes, ion-selective sensors, electonic tongue electrodes...) are also logarithmic (that usually implies reversible binding of analyte to the sensor). 

The present estimation of LoD is incorrect. You should take into account not only the point estimate (formula 3)  but also the regression variance of slope and intercept (to estimate these you have to fit dependence of lg(C) on intensity and build confidence intervals for slope and intercept). In other words, you need to calculate the confidence band for you calibration curve and the practical LOD will be the intersection of the upper confidence margin with the axis, not the intersection with the curve itself. Rouhly taking the point values from Fig. 5 my optimistic estimate of LoD yielded no better than 0.2 units of X axis units, that would be about 1.6 nM, which is much more reasonable than the presented estimate. Please consider and correct the value.

It is also highly recommended to do grammar checking, especially in the newly added parts.

Reviewer 3 Report

I appreciate the great efforts that the authors have made in response to the reviewer's questions and concerns. The revision clarifies almost all the points they raised and helps me (and hopefully readers) understand the current manuscript. Following please find the points I think the authors may still take into account.

- I am still not convinced that by using a 532 nm-laser excitation and a grating of 600 lines per mm you are working with a 1cm-1-spectral resolution. You wrote: "The typical measured spectral resolution of our Raman spectrometer has been found ~1 cm-1." At which conditions is it the case?

Table S1 should list all observed Raman bands. What about their relative intensities?

- "Point 13: Pg. 8, lns. 268-276: Can you specify from where - which structure - the COO- and NH2 vibrations/groups can be deduced? - Response: We are not able to infer the exact query raised by the reviewer. However, if it is related to the vibrational bands of sarcosine, then it can be clearly depicted from the molecular structure of sarcosine (C3H7NO2) there are so many possible vibrations of sarcosine molecule such as CH3 asymmetric stretching, CH2 stretching mode, NH2 deformation mode, COO¯ deformation mode etc. [1]." - I supposed the same regarding the structure of the sarcosine molecule. However, I expected some interaction mechanism explanations resulting in the observed spectral changes.

- "The typical observed standard deviation of the response has been found ~ 0.035 nm." - As I know, the standard deviation of the response is connected with the intensity of the signal. Then, why do you present it in nm?

- "Point 22: Pg. 10, lns. 332-338: Your calculated EF is ~2×107. You wrote: "…. the present technique is much more effective as compared to the work reported by Li et al. that uses the SHINERS method to get the EF of order ~107."

Why do you think your method is much more effective? Does the 2-times higher EF equal a much more effective method? - Response: We would like to thank the reviewer for this query. The method described by Li et al. is relatively tedious, and time-consuming as it owes a two-step synthesis technique. Sometimes, in the process of synthesizing the metal nanoparticles as well as their composites, the production of some undesired products or impurities deteriorate the performance of the materials. However, in our reported work we have used a simple Radio Frequency Sputtering method for the deposition of Ag nano islands along with the coating of silica nanospheres over it by spin coating method and we get equal EF to the work reported by Li et al. Thus, we believe that our reported technique is much more effective." - You should also incorporate this explanation into the manuscript.

Round 3

Reviewer 2 Report

The authors corrected the the specified issues.

The manuscript may be published.

Reviewer 3 Report

The authors responded: The typical observed spectral resolution of the Raman spectrometer has been found ~2.8 cm-1 when the Raman excitation wavelength is 532 nm, slit size is 10 microm and 600 l/mm, 500 nm blaze of the grating. This also corresponds to the difference between two successive data points of the obtained Raman spectrum.

Why is there still 1 cm-1-spectral resolution in the revised paper?

----

The authors should also indicate units for the relative intensities (e.g., a.u. or counts or % or...)
